# Prevalence and factors associated with moderate-to-severe anaemia among virally suppressed people with HIV at a tertiary hospital in Zambia

**Kingsley Kamvuma**[1,2]*, **Sepiso Masenga**[1], **Benson Hamooya**[1], **Warren Chanda**[1], **Sody Munsaka**[2]

**1** Department of Pathology and Microbiology, HAND Research Group, School of Medicine and Health Sciences, Mulungush University, Livingstone, Zambia, **2** Department of Biomedical Sciences, University of Zambia of Health Science and Medicine, Lusaka, Zambia

\* kamvumak@yahoo.com

**Data Availability Statement:** All relevant data are within the paper and its Supporting Information files.

## Abstract

### Objective

Anaemia is associated with an increased risk of disease progression and all-cause mortality among HIV-infected individuals, regardless of the type of anaemia, but the magnitude of the risk is greater with more severe forms of anaemia. Although anaemia PLWH has been extensively studied, the focus has primarily been on its prevalence and association with disease progression in untreated or poorly controlled HIV cases. This study aimed to investigate the prevalence, and factors associated with moderate-to-severe anaemia among virally suppressed HIV patients at a tertiary hospital in Zambia.

### Methods

We conducted a cross-sectional study of ART-treated PLWH for at least 6 months at Livingstone University Teaching Hospital (LUTH). Sociodemographic, clinical, and laboratory were the data collected. The primary outcome moderate to severe anaemia was defined as follows; moderate anemia as haemaoglobin levels between 8.0–10.9 g/ and severe anemia as haemoglobin levels less than 8.0 g/dL according to the WHO classification. Logistic regression was performed to identify factors associated with moderate-to-severe anaemia.

### Results

Among 823 participants with viral suppression, the overall prevalence of anaemia and moderate-to-severe anaemia was 29.4% (n = 242; 95% confidence interval (CI): 26.3–32.6) and 14.2% (n = 117, 95% CI: 11.7–18), respectively. In the adjusted logistic regression analysis, women had higher odds of moderate to severe anaemia compared to men (AOR 2.618, 95% CI 1.182–5.799). Lymphocyte count (AOR 0.525, 95% CI 0.31–0.90) and higher BMI (AOR 1.0671, 95% CI 1.01–1.13) were also significant factors. Microcytosis (AOR 49.79, 95% CI 12.95–191.49) and normocytosis (AOR 4.38, 95% CI 1.22–15.75) were strongly associated with higher odds compared to macrocytosis. NNRTI treatment was associated

**Funding:** The author(s) received no specific funding for this work.

**Competing interests:** The authors have declared that no competing interests exist.

with higher odds of anaemia compared to INSTI treatment (AOR 5.231, 95% CI 1.04–26.33). Traditional risk factors for anaemia like CD4+ count and tuberculosis infection were not significant.

## Conclusion

We found a higher prevalence of anaemia and moderate-to-severe anaemia in virally suppressed PLWH, suggesting factors beyond HIV contribute to the persistence of anaemia in this cohort. Women, lower lymphocyte count, higher BMI, low mean corpuscular volume (microcytosis) indicative of microcytic anaemia, and NNRTI-based ART regimens were independently associated with moderate-to-severe anaemia. Further research is warranted to explain the underlying mechanisms and optimize clinical management to improve outcomes among virally suppressed PLWH.

## Introduction

In the past few decades, significant progress has been made in managing HIV infection, transforming it from a once-devastating disease to a chronic, controllable condition with the advent of effective combinational antiretroviral therapy (ART) [1]. With the implementation of the "95-95-95" strategy, there is a growing trend of People Living with HIV (PLWH) achieving viral suppression [2]. However, despite this progress, PLWH continue to face a numerous health challenges, among which anaemia stands out as a persistent and debilitating complication [3]. The prevalence of anaemia varies across clinical settings. Recent studies suggests that anaemia is associated with an increased risk of disease progression, a heightened inflammatory profile including immune reconstitution inflammatory syndrome (IRIS), incident tuberculosis and all-cause mortality among HIV-infected individuals, regardless of type of anaemia, and the magnitude of the risk is greater with more severe forms anaemia [4–6]. Understanding the prevalence and severity of anaemia among PLWH is crucial for optimizing patient care in this population.

The pathogenesis of anaemia in virally suppressed PLWH remains unclear. However, chronic inflammation, even in the absence of active viral replication, contributes to disrupted erythropoiesis [7, 8]. Inflammatory cytokines interfere with iron metabolism, impairing red blood cell production [9]. Additionally, malabsorption, chronic bleeding and micronutrient deficiencies particularly in limited resource settings may further contribute to anaemia persistence in PLWH.

As antiretroviral therapy prolongs the lives of PLWH, it becomes essential to gain a comprehensive understanding of the dynamic factors shaping their health needs. Although anaemia in PLWH has been extensively studied, the focus has primarily been on its prevalence and association with disease progression in untreated or poorly controlled HIV cases [3, 10]. However, a critical gap exists in understanding the factors contributing to moderate-to-severe anaemia specifically among virally suppressed HIV patients, who represent a growing segment of the HIV population due to improved access to effective ART regimens. Therefore, this study aimed to evaluate the prevalence, and factors associated with moderate to severe anaemia among virally suppressed people living with HIV.

## Materials and methods

### Study design and setting

This was a cross-sectional study among virally suppressed (viral load <1000 copies/ml) adult PLWH who had been on ART for at least 6 months or more, between 1st September 2023 and

26th February 2024 at Livingstone University Teaching Hospital (LUTH). LUTH provides ART services (laboratory, pharmacy, clinical evaluation, and counselling) and is the largest referral hospital located in Southern part of Zambia servicing over 2.4 million people. The hospital offers HIV treatment and management services to approximately 4000 PLWH annually.

## Eligibility criteria

Following informed consent, we recruited adult PLWH aged 18 years and above who had been receiving ART for ≥ 6 months. In this study, pregnant women, participants with a known history of excessive menstrual bleeding, disorders of haemoglobin synthesis including Sickle cell anaemia and thalassaemia or a malignant neoplasm were excluded. During the initial interview, participants were asked detailed questions regarding their existing conditions, and thorough review of their medical histories.

## Sample size

The sample size calculation was conducted using the single population proportion formula. Assumptions: True Population Proportion (P): Due to the absence of previous similar studies in the area, we assumed a proportion of 50%. Precision: We set the desired precision (margin of error) at 0.05. The confidence level was set at 99%. Given the total population of 3,880 patients receiving antiretroviral therapy (ART) at LUTH, we initially determined a base sample size of 578. To account for potential issues such as ineligibility after consent and non-response, we applied a 30% contingency. The final sample size, after including this contingency, was calculated to be 823 participants.

## Study variables and definitions

Sociodemographic data, including age, sex, blood pressure, and physical exercise were collected from participants and health records (SmartCare and patient files) using a structured questionnaire and data collection form. Trained research assistants administered the questionnaire and obtained blood samples. In case of clinically abnormal findings, participants were referred to a physician. Blood pressure (using a digital machine- Omron-HEM-7120, USA) was determined by calculating an average of three readings obtained after resting participants for 5 minutes and taking readings one minute apart. The height (cm), weight (kg) and waist circumference were measured using a height measurement chart, digital scale and tape measure, respectively.

## Operational definitions

Anemia was defined based on the World Health Organization (WHO) classification as a hemoglobin concentration lower than normal: <12 g/dL in women and <13 g/dL in men [11]. The type of anemias was assessed using mean cell volume (MCV) values, categorizing them as microcytosis (<80 fL), normocytosis (80–100 fL), and macrocytosis (>100 fL) [12]. Viral suppression was defined according to the Zambia Consolidated Guidelines for Treatment and Prevention of HIV Infection as consistent adherence to antiretroviral therapy (ART) and achieving a viral load of less than 1,000 copies of HIV RNA per milliliter of blood [12].

**Primary outcome.** The primary outcome moderate to severe anaemia was defined as moderate anemia as haemaoglobin levels between 8.0–10.9 g/ and severe anemia as haemoglobin levels less than 8.0 g/dL according to the WHO [11].

## Blood samples and measurements

We collected blood samples to measure viral load and CD4+ count in ethylenediaminetetra-acetic acid (EDTA) containers., while the viral load was analysed using Ampliprep/Taqman 96 PCR analyser which has a lower limit of detection of 20 copies/mL. Haemoglobin (Hgb) values were determined using the hematology analyzer Sysmex XT2000 (Abbott Laboratories Diagnostics Division, USA) and CD4+ T cells were assayed using the BD FACSCOUNT system (Becton Dickenson and Company, California, USA).

## ART regimens

Non-nucleoside reverse transcriptase inhibitor (NNRTI) regimens contained efavirenz (EFV) or nevirapine (NVP) with one of the following nucleoside reverse transcriptase inhibitors (NRTIs): Abacavir and lamivudine/emtricitabine (ABC/XTC) or tenofovir disoproxil fumarate and lamivudine/emtricitabine (TDF/XTC). Protease inhibitor (PI) regimens contained either lopinavir/ritonavir (LPV/r) or atazanavir/ritonavir (ATV/r) with one of the following NRTI combinations: ABC/XTC or zidovudine/XTC (AZT/XTC) or TDF/XTC. An integrase strand transfer inhibitor (INSTI) regimen contained Dolutegravir (DTG) with TDF/lamivudine (TDF/3TC).

## Statistical analysis

All statistical analyses were conducted using SPSS software version 22. Categorical data were summarized using frequencies and proportions, while continuous variables were summarized using medians and interquartile ranges (IQR) due to non-normal distribution, which was confirmed using Q-Q plots and the Shapiro-Wilk test. The Pearson chi-square test was employed to assess statistically significant associations between categorical variables. Mann–Whitney U test (for two unmatched groups) was used to ascertain a statistical difference between two medians. Logistic regression (univariable and multivariable) was utilized to estimate factors associated with moderate to severe anaemia. Covariates included in the final model were selected based on published evidence and variables found to be statistically significant in univariable analysis. Model fitness was assessed using the Hosmer-Lemeshow goodness-of-fit test, with statistical significance defined as $p < .05$.

## Ethical considerations

Ethical approval for the study was obtained on 7th August 2023 from the University of Zambia Biomedical Research Ethics Committee (UNZABREC- REF. REF. NO. 4062–2023). Data was collected between 1st October 2023 and 26th February 2024 at Livingstone University Teaching Hospital (LUTH). The purpose of the study was explained to all the participants in a language familiar to them, and they provided written informed consent after agreeing to take part in the study. Confidentiality and anonymity was maintained during data collection by ensuring that the data were de-identified.

## Reporting format

We adhered to the strengthening the reporting of observational studies in epidemiology (STROBE) guidelines for reporting observational studies. See S1 File for details.

# Results

## Sociodemographic and clinical characteristics of the study participants

The study included 823 individuals undergoing ART treatment who had achieved viral suppression. Among them, there was a predominance of women, accounting for 61% (n = 502) of

the participants. The median age of the participants was 41 years (interquartile range (IQR) 40,54), and the age range (18–79) years. The overall prevalence of anaemia and moderate-to-severe anaemia was 29.4% (n = 242; 95% confidence interval (CI): 26.3–32.6) and 14.2% (n = 117 95% CI: 11.7–18) respectively. With respect to ART regimens, majority were on NNRTI-based regimens (81%, 660/823, followed by INSTI-based regimens (14.5%, n = 118), and PI-based regimens (4.7%, n = 38). Furthermore, among the participants, 0.8% (n = 4) had active tuberculosis, 6.7% (n = 51) were positive for Hepatitis B surface antigens, and 14.4% (n = 136) had syphilis infections.

## Relationship of anaemia status with demographic and clinical factors

Table 1 shows an analysis of factors associated with moderate-to-severe anaemia. A higher prevalence of moderate-to-severe anaemia observed among women compared to men (19.7% vs. 5.6%, p = 0.001). Additionally, ART regimen was associated with moderate-to-severe anaemia, specifically individuals on non-nucleoside/nucleotide reverse transcriptase inhibitor (NNRTI)-based regimens showing a higher prevalence of moderate-to-severe anaemia compared to those on integrase strand transfer inhibitor (INSTI) and protease inhibitor-based regimens (16.2% vs. 5.9% and 5.3% respectively, p = 0.004). Laboratory findings also reveal significant associations, with parameters such as low mean corpuscular volume (MCV) indicative of Microcytosis and Low lymphocyte count strongly correlated with moderate-to-severe anaemia (p < 0.05). Other variables like physical exercise, and comorbidities such as Hepatitis B Surface antigen, hypertension, and tuberculosis infection show no significant associations with moderate-to-severe anaemia among our study participants.

## Relationship between study variables and outcome status (moderate-to-severe Anaemia)

The graphs illustrate the distribution of median values stratified by the outcome status in our study cohorts. Clinical variables such as Duration on ART in months, mean corpuscular volume, and absolute lymphocyte count levels showed significant differences when their medians were compared with outcome variable moderate-to-severe anaemia. BMI, CD4+ count and Viral load levels did not show significant difference between the two groups (see **Fig 1**). Fig 2 shows the prevalence of moderate to severe anaemia across sex, ART regimen, type of anaemia, and physical exercise engagement among PLWH. Prevalence of anaemia was significantly higher among women participants, those with RBC microcytosis based on MCV values and individuals on NNRTI P<0.05.

## Logistic regression analysis of factors associated with moderate-to-severe anaemia

Table 2 shows the logistic regression analysis of factors associated with moderate-to-severe anaemia. At unadjusted analysis, women were found to have four times higher odds of having moderate-to-severe anaemia than men (OR 4.14, 95% CI 2.45–6.98, p < 0.001). Patients undergoing NNRTI treatments faced higher odds compared to those on INSTI treatments (OR 3.07, 95% CI 1.39–6.77, p = 0.005). Microcytic and normocytic anemia, as indicated by mean corpuscular volume (MCV), were significantly associated with higher odds of anemia compared to macrocytic anemia, with odds ratios (OR) of 45.14 (95% CI: 14.88–136.9, p < 0.001) and 3.51 (95% CI: 1.22–10.10, p = 0.024), respectively. A lower lymphocyte count was associated with higher odds of moderate to severe anemia in HIV (OR 0.59, 95% CI 0.39–0.88, p = 0.009). Being on ART treatment for more than 4 years was associated with higher

**Table 1. Shows the sociodemographic, and clinical factors according to anaemia status.**

| Characteristics | | Moderate to severe anaemia, n (%) | | |
|---|---|---|---|---|
| | Total (823) | Yes (14.2%, n = 117) | No (85.8%, n = 706) | p-value |
| Age in years, m | 823 | 42 (40, 48) | 41 (40, 54) | 0.429 |
| **Sex** | | | | **0.001** |
| *Men* | 321 | 18 (5.6) | 303 (94.4) | |
| *Women* | 502 | 99 (19.7) | 403 (80.3) | |
| **Regimen** | | | | **0.004** |
| *INSTI* | 118 | 7 (5.9) | 111 (94.1) | |
| *PI* | 39 | 2 (5.3) | 36 (94.7) | |
| *NNRTI* | 667 | 107 (16.2) | 553 (83.8) | |
| **HBsAg** | | | | 0.177 |
| *Yes* | 51 | 4 (7.8) | 47 (92.2) | |
| *No* | 705 | 47 (6.7) | 658 (93.3) | |
| *Missing* | 67 | - | - | |
| **Hypertension** | | | | 0.482 |
| *Yes* | 76 | 8 (10.5) | 68 (89.5) | |
| *No* | 447 | 68 (15.2) | 379 (74.8) | |
| *Missing* | 300 | - | - | |
| **TB** | | | | 0.403 |
| *Yes* | 4 | 0 (0) | 4 (100) | |
| *No* | 514 | 76 (14.8) | 438 (75.2) | |
| *Missing* | 305 | - | - | |
| **Syphilis** | | | | 0.152 |
| *Yes* | 136 | 14 (10.3) | 122 (89.7) | |
| *No* | 687 | 103 (14.9) | 584 (75.1) | |
| **Types of anaemia** | | | | **<0.001** |
| *Microcytois* | 66 | 39 (59.1) | 27 (40.9) | |
| *Normocytosis* | 317 | 32 (8.1) | 285 (91.9) | |
| *Macrocytosis* | 129 | 4 (3.1) | 125 (96.9) | |
| *Missing* | 311 | - | - | |
| **Duration ART mo, m(IQR)** | | | | **0.019** |
| *Year 1* | 72 | 7 (9.7) | 65 (90.3) | |
| *Year 2* | 95 | 6 (6.3) | 89 (93.7) | |
| *Year 3* | 50 | 9 (18) | 41 (82) | |
| *4 years and more* | 297 | 54 (18.2) | 243 (81.8) | |
| *Missing* | 309 | - | - | |
| **CD4+ t (cells/mL) m(IQR)** | 823 | 492 (461, 877) | 518 (372, 877) | 0.310 |
| **BMI kg/m2,m(IQR)** | 823 | 22.7 (4.7, 5.6) | 21.8 (4.3, 5.3) | 0.74 |
| **Viral load Copies/uL** | 823 | 20 (0.0, 27) | 20 (0.0, 47) | 0.160 |
| **Lymphocytes(109 cells/L), m(IQR** | 488 | 1.86 (1.5, 2.0) | 2.12 (1.6, 2.6) | **0.002** |
| *missing* | 335 | - | - | |
| **Monocytes (109 cells/L), m(IQR)** | 488 | 0.32 (0.21, 0.51) | 0.38 (0.29, 0.50) | 0.986 |

(*Continued*)

**Table 1.** (Continued)

| Characteristics | Total (823) | Moderate to severe anaemia, n (%) | | |
| --- | --- | --- | --- | --- |
| | | Yes (14.2%, n = 117) | No (85.8%, n = 706) | p-value |
| *Missing* | 335 | - | - | |

Values are presented as median (m) and Interquartile range (IQR) ±standard deviation for the continuous variables and frequency (percentage) for the categorical variables. mo: months

eGFR: estimated glomerular filtration rate, MCV: mean corpuscular volume, ALT: Alanine aminotransferase, TB: Tuberculosis, HBsAg: Hepatitis B Surface antigen FBS: Fasting blood sugar, VL: Viral load, ESR: erythrocyte sedimentation rate, NNRTI = non-nucleoside/nucleotide reverse transcriptase inhibitor (EFV = efavirenz and NVP = Nevirapine), PI = protease nhibitor (LPV/r = lopinavir/ritonavir and ATV/r = atazanavir/ritonavir), INSTI = integrase strand transfer inhibitor (DTG = dolutegravir), NRTI = nucleotide reverse transcriptase inhibitor.

odds of moderate to severe anemia in this study cohort (OR 2.95, 95% CI 1.46–5.93, p = 0.002).

## Discussion

The study reveals that 29.4% and 14.2% of virally suppressed individuals with HIV who are on ART had anaemia and moderate-to-severe anaemia, respectively. This indicates that, despite effective viral control, anaemia continues to be a prevalent issue, particularly affecting women. The findings are consistent with other research, which shows a broad range of anaemia prevalence, from 7% to 95%, among those treated with ART depending on the clinical context and geographic setting [3, 13]. A study from China showed a decreasing trend in anaemia prevalence over time among a large cohort after initiating ART, yet our study show that the prevalence remain significantly high. Theirs study show a prevalence of anaemia among 436,658

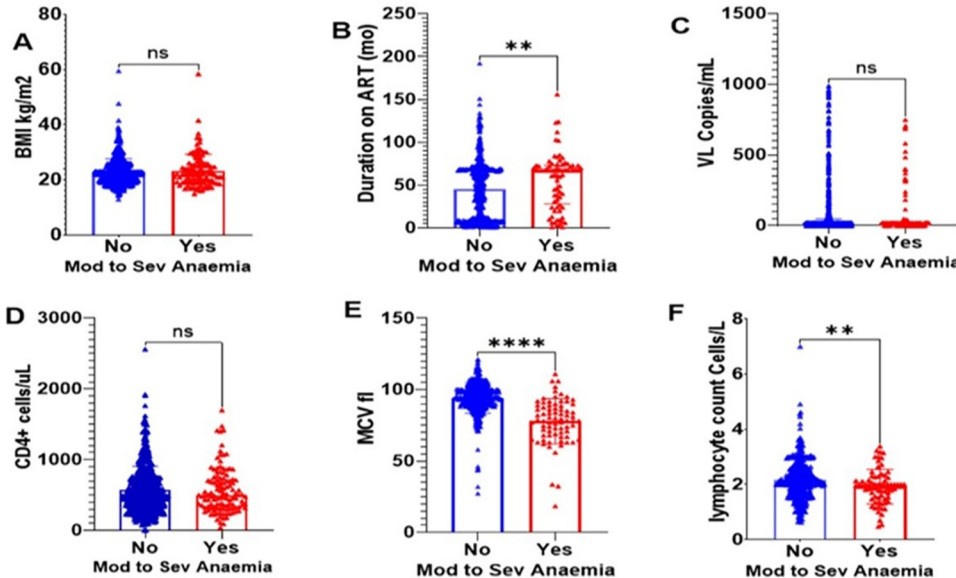

**Fig 1. A to F: shows the stratification of clinical variables by anaemia status.** BMI Body Mass Index, VL Viral Load, ART Antiretroviral therapy, MCV Mean Corpuscular Volume, ns not significant, ** P<0.005, **** P<0.0001.

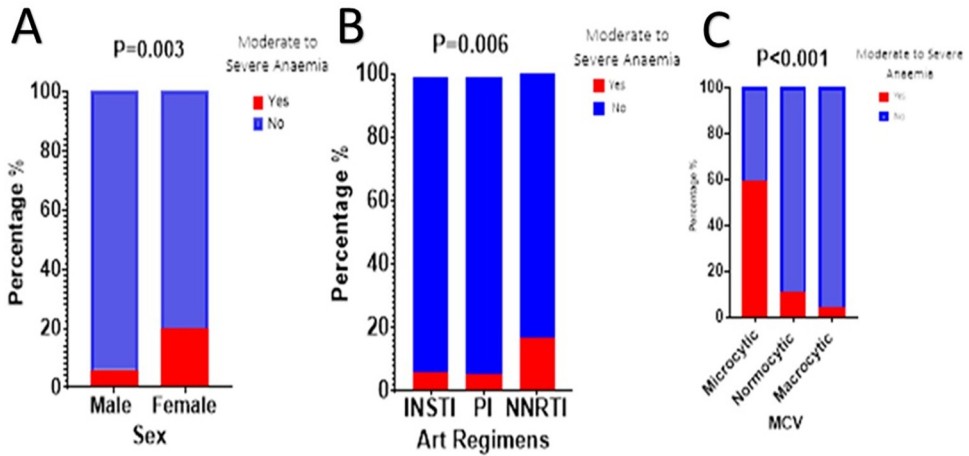

**Fig 2.** A to C: illustrates the prevalence of Moderate to Severe Anaemia based on Sex (A), ART regiment (B), Type of Anaemia (C). PI Protease Inhibitor, INSTI Integrase Strand Transfer Inhibitor, NNRTI Non Nucleotide Reverse Transcriptase Inhibitor, MCV Mean Corpuscular Volume.

**Table 2. Factors associated with moderate to severe anaemia.**

| Variables | Unadjusted Odds ratio | | p- value | Adjusted Odds | | p- value |
|---|---|---|---|---|---|---|
| | OR | 95%CI | | AOR | 95%CI | |
| **Age** | 0.99 | 0.97–1.01 | 0.255 | 1.012 | 0.98–1.04 | 0.440 |
| **Sex**, Women vs Men | 4.14 | 2.45–6.98 | **<0.001** | 2.618 | 1.182–5.799 | **0.018** |
| **BMI** (kg/m2) | 1.04 | 0.98–1.05 | 0.439 | 1.067 | 1.01–1.13 | **0.016** |
| **Regimen** | | | | | | |
| Insti (Dtg) | | | 1 | | | |
| PI (LPV/R&ATZ/R) | 0.88 | 0.18–4.43 | 0.878 | 2.458 | 0.32–18.86 | 0.387 |
| Nnrti (Efv&Nvp) | 3.07 | 1.39–6.77 | **0.005** | 5.231 | 1.04–26.33 | **0.045** |
| **Duration on ART** | | | | | | |
| Year 1 | | | 1 | | | |
| Year 2 | 3.42 | 1.49–0.66 | 0.339 | 2.011 | 0.53–7.69 | 0.307 |
| Year 3 | 1.78 | 0.95–3.35 | 0.074 | 1.702 | 0.50–5.75 | 0.392 |
| >4yrs | 2.95 | 1.46–5.93 | **0.002** | 0.595 | 0.15–2.34 | 0.457 |
| **Types of Anaemia** | | | | | | |
| *Macrocytic* | | | 1 | | | |
| *Normocytic* | 3.51 | 1.22–10.10 | **0.024** | 4.38 | 1.22–15.75 | **0.024** |
| *Microcytic* | 45.14 | 14.88–136.9 | **<0.001** | 49.79 | 12.95–191.49 | **<0.001** |
| **CD4 Count** (cell/ul) | 1.00 | 0.99–1.00 | 0.667 | 1.00 | 0.99–1.00 | 0.983 |
| **Lymphocytes** | 0.59 | 0.39–0.88 | **0.009** | 0.525 | 0.31–0.90 | **0.017** |
| **Viral load** | 1.00 | 0.99–1.00 | 0.619 | | | |
| **Tuberculosis**, yes | 1.08 | 0.65–1.78 | 0.773 | | | |
| **HBsAg**, yes | 2.02 | 0.71–5.71 | 0.186 | | | |
| **Syphilis**, yes | 1.54 | 0.85–2.78 | 0.154 | | | |

At adjusted analysis, women had a significantly higher odds of moderate-to-severe anemia than men (AOR 2.65, 95% CI 1.21–5.81, p = 0.015). A lower lymphocyte count was associated with higher odds (AOR 0.54, 95% CI 0.33–0.90, p = 0.019) of moderate to severe anemia. Higher Body Mass Index (BMI) also increased the odds, albeit slightly, with an AOR of 1.062 (95% CI 1.01–1.12, p = 0.018). Microcytic anemia (adjusted OR 49.79, 95% CI 12.95–191.49, p < 0.001) and normocytic anemia (OR 4.38, 95% CI 1.22–15.75, p = 0.024) were significantly associated with higher odds of moderate to severe anemia. NNRTI treatment (adjusted OR 5.231, 95% CI 1.04–26.33, p = 0.045) remained strongly associated with higher odds of moderate to severe anemia. Notably, traditional risk factors like CD4+ count and tuberculosis infection were not significant in this study cohort.

PLWH initiating ART was 29%, and annually, this prevalence reduced to 17.0%, 14.1%, 13.4%, 12.6%, and 12.7%, respectively [14]. The prevalence observed in this study was slightly lower than the prevalence reported in the Ethiopian study (33%) among women, which specifically focused on PLWH on ART. Both studies, however, did not report on the virologic status of their participants [15]. Our emphasizes the ongoing need to address factors contributing to anaemia, such as nutritional deficiencies, persistent low-grade inflammation, and ART-related toxicities, to improve clinical outcomes for PLWH. Further research is necessary to explain the evolving epidemiology of anaemia in the context of HIV infection and to evaluate the impact of ART on anaemia.

The observation of women having significantly higher odds of moderate-to-severe anaemia compared to men aligns with previous studies in HIV populations. Women are most likely to experience iron deficiency and subsequent anaemia. Women's sex hormones may influence erythropoiesis and iron utilization [10, 16].

The inverse relationship between lymphocyte count and moderate-to-severe anaemia suggests that immune status remains a crucial factor in the health of PLWH, even when viral suppression is achieved. Lymphocyte depletion could contribute to anaemia through various mechanisms, including dysregulated cytokine signalling leading to impaired iron metabolism [17]. Surprisingly, CD4+ count did not show significant associations but effective antiretroviral therapy (ART) and viral suppression may lead to CD4+ cell count recovery [18].

The positive association between higher BMI and moderate-to-severe anaemia is to a certain extent contradictory to expectations, as obesity is often associated with increased iron stores [19]. However, Elevated BMI is linked to chronic inflammation and altered adipokine secretion, affecting erythropoiesis [20]. Furthermore, comorbid conditions commonly associated with obesity, such as chronic kidney disease, could also contribute to anaemia risk. Addressing obesity-related inflammation and iron status is crucial.

Our study found a strong association between microcytic anaemia and moderate-to-severe anaemia when compared to macrocytic anaemia, suggesting a pattern consistent with a classic presentation of iron-deficiency anemia and anaemia chronic disease [21]. Factors such as inadequate dietary intake, malabsorption, chronic bleeding and chronic inflammation may exacerbate iron deficiency anaemia among PLWH in limited settings.

NNRTIs were associated with higher odds of moderate-to-severe anaemia compared to INSTIs (DTG-based regimen). Some NNRTIs may suppress bone marrow function, affecting erythropoiesis but the exact mechanisms underlying this association are not fully elucidated [22]. However, NNRTIs have been implicated in bone marrow suppression due to ART-associated mitochondrial toxicity, which could contribute to decreased erythropoiesis [23, 24]. Long-term effects, including the potential for haematological complications, must be carefully considered.

This study has several limitations, the study did not account for other possible causes of anaemia, such as nutritional factors (serum iron, folate and vitamin B12) and inflammation. Furthermore, Our study had missing data due to challenges with laboratory analysis and incomplete patient information in the smart care system, but we determined that the data were Missing Completely at Random (MCAR), indicating no systematic differences between missing and observed data. However, the missingness of the data may have impacted the inference of our analysis. We also acknowledge that it may not be adequately powered to fully assess other secondary association. Strengths of the study include its robust methodology, which employed standardized data collection tools and laboratory measurements. The inclusion of a large sample size adds to the study's statistical power and allows for precise estimation. The participants in this cohort had achieved virologic suppression, below 1000 cells/uL, mitigating the potential effects of virologic failure.

The findings from this study have significant clinical implications for PLWH. Firstly, the observed prevalence of anaemia and moderate-to-severe anaemia highlights the need for clinicians to pay attention to hematological parameters in this population, even when viral load is well controlled. Secondly, the identification of risk factors such as women gender, low lymphocyte count, high BMI, microcytic anaemia and specific antiretroviral therapy (ART) regimens provides valuable insights for health professionals and policy makers to tailor interventions accordingly. By addressing these factors, healthcare providers can optimize anaemia management, potentially improving patient outcomes, quality of life, and overall health.

## Conclusion

These findings expand upon existing literature on anaemia in HIV-infected populations and adds knowledge to the evolving dynamics of PLWH on treatment. We found a higher prevalence of anaemia (29%) and moderate-to-severe anaemia (14.2%) in virally suppressed PLWH, suggesting that factors beyond HIV contribute to the persistence of anaemia in our cohort; and it was significantly associated with sex, immunological, nutritional, and ART factors. We found that Women, lower lymphocyte count, higher BMI, low mean corpuscular volume indicative of microcytic anaemia, and NNRTI-based ART regimens were independently associated with moderate-to-severe anaemia. Understanding these factors allows healthcare providers to proactively manage and prevent moderate-to-severe anaemia among PLWH. There is a need to conduct longitudinal studies to understand the risk factors of moderate-to-severe among virally suppressed PLWH on ART.

## Supporting information

**S1 File. The strobe checklist.**
(PDF)

**S2 File. Minimal dataset.**
(XLSX)

## Author Contributions

**Conceptualization:** Kingsley Kamvuma, Benson Hamooya, Sody Munsaka.

**Formal analysis:** Kingsley Kamvuma, Sepiso Masenga, Benson Hamooya.

**Investigation:** Kingsley Kamvuma.

**Methodology:** Kingsley Kamvuma, Sepiso Masenga, Sody Munsaka.

**Project administration:** Kingsley Kamvuma.

**Software:** Sepiso Masenga.

**Supervision:** Sepiso Masenga, Benson Hamooya, Sody Munsaka.

**Visualization:** Kingsley Kamvuma, Warren Chanda, Sody Munsaka.

**Writing – original draft:** Kingsley Kamvuma.

**Writing – review & editing:** Sepiso Masenga, Benson Hamooya, Warren Chanda, Sody Munsaka.

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
