## [Decision Letter · Decision Letter 0]

11 Jun 2024

PONE-D-24-17294Prevalence and Factors Associated with Moderate-to-Severe Anaemia Among Virally Suppressed People with HIV at a Tertiary Hospital in ZambiaPLOS ONE

Dear Dr. Kamvuma,

Thank you for submitting your manuscript to PLOS ONE. After careful consideration, we feel that it has merit but does not fully meet PLOS ONE’s publication criteria as it currently stands. Therefore, we invite you to submit a revised version of the manuscript that addresses the points raised during the review process.

We look forward to receiving your revised manuscript.

Kind regards,

Nitin Gupta, MBBS, MD, DM, AAHIVS, DTM&H

Academic Editor

PLOS ONE

Journal Requirements:

Reviewers' comments:

Reviewer's Responses to Questions

**Comments to the Author**

1. Is the manuscript technically sound, and do the data support the conclusions?

Reviewer #1: Yes

Reviewer #2: Partly

2. Has the statistical analysis been performed appropriately and rigorously? 

Reviewer #1: Yes

Reviewer #2: No

3. Have the authors made all data underlying the findings in their manuscript fully available?

Reviewer #1: No

Reviewer #2: No

4. Is the manuscript presented in an intelligible fashion and written in standard English?

Reviewer #1: Yes

Reviewer #2: Yes

5. Review Comments to the Author

Reviewer #1: Thank you for performing this study. While we read the same with great interest, a few queries came to mind.

1. //In this study, pregnant women, participants with a known history of excessive menstrual bleeding, disorders of haemoglobin synthesis including Sickle cell anaemia and thalassaemia or a malignant neoplasm were excluded//- How was this exclusion approched? On the basis of history or were blood investigations performed to establish the diagnosis?

2. How were clinically abnormal findings defined? Were the research assistants trained in clinical examination?

3. Moderate and severe anemia needs to be clearly defined, instead of a blanket value of less than 10.9 g % alone.

4. In Table 1. The association with normocytosis, macrocytosis and eosinopil percentages with anemia have not been depicted. Was this looked into?

5. //While being on ART treatment for more than 4 years was associated with higher odds

of moderate to severe anaemia in this study cohort.//- This statement would have more value when combined with the break up of types of ART used.

6. //Our emphasizes the ongoing need to address factors contributing to anaemia, such as

nutritional deficiencies, persistent low-grade inflammation, and ART related toxicities, to

improve clinical outcomes for PLWH.// While study was designed, were attempts made to discuss possible methods to assess low grade ongoing inflammation?

7. //Female sex hormones may influence erythropoiesis and iron utilization// Would be interesting to look at status of anemia(mild vs moderate vs severe) in pre and post menopausal women.

8. //Factors such as inadequate dietary intake, malabsorption, chronic bleeding and chronic inflammation

may exacerbate iron deficiency anaemia among PLWH in limited settings// This as well as previous assessment of obesity and inflammation, suggests the need for looking at possible pointers of inflammation. It would also be important to look at serum ferritin to classify the IDA vs anemia of chronic disease.

9.It would be important to review the questionnaire that was initially used by the research assistants. Was this questionnaire validated and what was the language it was administered in?

10. Since the study's outcome variable was anemia, the study may not be adequately powered to assess the other associations that have been unconvered.

Reviewer #2: The authors present a cross-sectional study to examine factors associated with anemia among people living with HIV at a single tertiary medical center in Livingstone, Zambia. This is an important topic, especially as Zambia approaches population level epidemic control; however, there are several areas the authors may wish to consider.

Introduction

1) minor comment - unclear what is meant by "latently PLWH". The study appears to include only people HIV who are on ART, whereas latent infection would imply a natural history study of people who have seroconverted but had not yet initiated ART or begun to exhibit symptoms consistent with immunologic suppression.

2) minor comment - recommend removing the term "sub-optimal chronic inflammation from the second paragraph. Agreed that chronic inflammation is sub-optimal but the placement of the word implies there could be more optimal chronic inflammation.

Methods

1) minor-comment suggest rephrasing the last sentence in study design and setting, would recommend stating first that LUTH serves over 2.4 million services and among general and specialty in patient and out patient services it also offers ART services, or state how many people with HIV LUTH serves annually.

2) the sample size calculation is unclear. It appears the authors used a sample size calculation for a population survey with a design effect of 1 and 1 cluster; however, the 30% contingency is unclear. As there generally isn't loss to follow-up in a cross-sectional study, is this to account for drop out? or ineligibility after consent? if so it would appear the sample size should have been 1,176 to make up for drop out. More detail here would be useful.

3) There may be a typo in the study variable section. Anaemia is defined twice. The first appears to be defining RBC development, macrocytosis for example is associated with levels of anemia but can also be associated with B-12 deficiency, hypothyroidism and HIV itself. You can have mirocytosis with or without anaemia. A clearer description of what this measuring defining would be useful.

4) minor-comment - in blood samples and measurements, I think you mean you collected blood samples to measure viral load and CD4 cell count.

5) minor comment - would be useful to note the lower limit of detection for the Taqman 96

6) minor-comment - the authors appear to have reported the procedure for CD4 cell counts twice.

7) please report which biochemical analyses were done on the Pentra and which were done on the HumaStar

Results

1) The proportion of patients on NNRTI-based regimens appears exceptionally high especially considering the 2020 national ART guidelines recommended patients with a suppressed viral load on an NNRTI containing regimen should be switched to DTG. The authors may want to check whether they abstracted the patients initial ART regimen at ART initiation or their current regimen.

2) in the first paragraph on page 8 the authors report people in NNRTI's had a higher prevalence of moderate-to-severe anemia compared to those in INSTI but then report 3 prevalence values - 16.2 vs 5.9 and 5.3 respectively, what is the third value referring to?

3) The authors may wish to consider including a row for missing for each of the factors in table 1 (but not include them in the chi-square comparison). If there are 823 total subjects in this study, it appears there are variables where they do not have information on all participants (i.e., HBsAG, Hypertension, TB, Microcytosis, duration of ART, lymphocytes, and monocytes). The comparison of microsytosis yes no is a bit confusing. The reader should have the opportunity to independently consider the potential impact of the missing data on inferences.

4) It is unclear why the authors investigated microcytosis as a binary variable rather than the three levels of MCV listed in the methods.

5) 38% of the study subjects appear to be missing and MCV values, and no data is presented on the proportion with normocytosis or macrocytosis, yet the authors report that the prevalence of anaemia was significant higher for those with microcytosis. The level of missing data and its potential impact on inferences should not be ignored. simiarly, if no subjects had MCV values consistent with normo or macrocytosis that should be reported.

6) minor comment, the p for p-value should be lowercase.

7) in looking at figure 1, did the authors explore outliers or implausible values? Also related to the previous comment on MCV volume, based on panel E in figure 1 there are a number of subjects who would have been classified as normo or macrocytotic. The treatment of this measurement needs better explanation.

9) Recommend the authors use the language women rather than females since they refer to men rather than males throughout the text.

10) please report the 95% confidence interval along with the OR in the text of the results.

11) As a cross-sectional study it would be more accurate to report higher odds than higher chance and refrain from using the term "developing anaemia". Temporality cannot be established with this study design and therefore the authors have no way of knowing whether the subject developed anaemia after starting an NNRTI-based regimen or whether they had it before they started ART. Also, per previous comment, please confirm that the NNRTI-based regimen was current at the time of the blood draw or whether that was their first regimen when starting treatment.

12) The authors appear to have used all continuous variables as continuous in the logistic regression model; however, it is unclear whether they tested the assumption of a linear association with the outcome prior to maintaining it as a continuous regimen. Would recommend that the authors explore whether age, bmi, CD4 cell count, and lymphocyes are linearly associated with the outcome.

13) The inclusion of viral load in the model is confusing as the authors indicated that to be eligible for the study they had to have a suppressed viral load <1000 copies/mL. What is the range of values in the viral load variable? How did the authors treat viral load for those whose VL was below the lower limit of detection of the test. Suggest removing viral load from the model.

Discussion:

1) until some of the issues raised are addressed it is difficult to review the discussion. There is additionally no mention of missing data as limitation of the study.

6. PLOS authors have the option to publish the peer review history of their article (what does this mean?). If published, this will include your full peer review and any attached files.

Reviewer #1: **Yes: **Dr. Kutty Sharada Vinod

Reviewer #2: No

---

## [Author Response · Author response to Decision Letter 0]

28 Jun 2024

We appreciate the valuable feedback and insightful comments provided by you and the reviewers on our manuscript titled “Prevalence and Factors Associated with Moderate-to-Severe Anaemia Among Virally Suppressed People with HIV at a Tertiary Hospital in Zambia” (PONE-D-24-17294). We have carefully considered each comment and have made the necessary revisions to address them. Below, we provide a detailed response to each point raised.

Reviewer No. 1

1. How was the exclusion of pregnant women, participants with a known history of excessive menstrual bleeding, disorders of haemoglobin synthesis including Sickle cell anaemia and thalassaemia, or a malignant neoplasm approached? Were these exclusions based on history, or were blood investigations performed to establish the diagnosis?

The exclusion of participants with a known history of excessive menstrual bleeding, disorders of haemoglobin synthesis (including Sickle cell anaemia and thalassaemia), and malignant neoplasms was based on history and reviews of their medical records. During the initial interview, participants were asked detailed questions regarding their existing conditions, and thorough review of their medical histories. We did not perform any blood to establish these diagnoses. See lines 90-91

2. How were clinically abnormal findings defined? Were the research assistants trained in clinical examination?

The research assistants involved in this study were clinicians and nurses with extensive experience in clinical examination and patient care. Additionally, they underwent specific training for this study to ensure standardized data collection procedures. Furthermore, clinically abnormal findings were defined based on standard medical guidelines and reference ranges for various clinical parameters, including blood pressure, haemoglobin levels, and other relevant laboratory values. 

3. Moderate and severe anemia needs to be clearly defined, instead of a blanket value of less than 10.9 g % alone.

We appreciate the comment and understand the need for clarity in defining moderate and severe anemia. In our study, we followed the World Health Organization's (WHO) guidelines for classifying anemia based on hemoglobin (Hb) concentrations. We have revised the definition to:

Moderate anemia as Hb levels between 8.0-10.9 g/dL (according to WHO, this corresponds to a Hb level of 8.0-10.9 g/dL for adults) and Severe anemia as Hb levels less than 8.0 g/dL (according to WHO, this corresponds to a Hb level of <8.0 g/dL for adults). See lines 118-131 under operational definitions

4. In Table 1. The association with normocytosis, macrocytosis and eosinopil percentages with anemia have not been depicted. Was this looked into?

We have re-analyzed the data and included the results in the revised table 1. However, we did not collect data on eosinophil percentages in our dataset, and therefore, we are unable to include it in our analysis. We appreciate your feedback and the opportunity to revise our manuscript

5. //While being on ART treatment for more than 4 years was associated with higher odds of moderate to severe anaemia in this study cohort.//- This statement would have more value when combined with the break up of types of ART used.

Thank you so much for the question. Most of our participants were on NNRTI-based regimens (81%). However, the result was only significant at bivariate or unadjusted analysis and non-significant at adjusted analysis. In our model, we have also accounted for the ART regimens the participants were receiving. We have provided a summary table of ART regimens for context but have not included a breakdown by duration to maintain focus on the anemia outcomes.

 year 1 year 2 year 3 year 4 P value

ART regimen < 0.001

PI 38 21 (55.3) 6 (15.8) 1 (2.6) 10(26.3) 

NNRTI 351 14 (4) 22 (6.3) 29 (8.3) 286 (81.5) 

INSTI 118 117 (99.2) 0 (0) 0 (0) 1(0.8) 

6. Our emphasizes the ongoing need to address f actors contributing to anaemia, such as nutritional deficiencies, persistent low-grade inflammation, and ART related toxicities, to improve clinical outcomes for PLWH.// While study was designed, were attempts made to discuss possible methods to assess low grade ongoing inflammation?

Thank you for your insightful comments regarding the assessment of low-grade inflammation and nutritional deficiencies in our study. While specific nutrient levels and detailed inflammatory markers were not measured, we utilized BMI as indicator for nutritional status and lymphocyte and monocyte counts as proxies for inflammatory processes. Our study also focused on analyzing the impact of different ART regimens and their duration on anemia, highlighting the significant influence of ART-related toxicities. We acknowledge the need for future studies to include direct measurements of specific nutrients and inflammatory markers to further elucidate their contributions to anemia among PLWH.

7. Female sex hormones may influence erythropoiesis and iron utilization// Would be interesting to look at status of anemia(mild vs moderate vs severe) in pre and post menopausal women.

Thank you for your insightful comment regarding the potential influence of female sex hormones on erythropoiesis and iron utilization. We agree that analyzing the status of anemia (mild, moderate, severe) in pre- and post-menopausal women would be valuable. However, we did not collect data on menopausal status among our study participants. Interestingly, strictly based on age (with a cutoff at 50 years), premenopausal women were at a higher risk of anemia but we recognise that women reach menopause at different age ranges. Future studies should include menopausal status to explore these hormonal influences more comprehensively.

8. Factors such as inadequate dietary intake, malabsorption, chronic bleeding and chronic inflammation may exacerbate iron deficiency anaemia among PLWH in limited settings// This as well as previous assessment of obesity and inflammation, suggests the need for looking at possible pointers of inflammation. It would also be important to look at serum ferritin to classify the IDA vs anemia of chronic disease.

Thank you for your valuable comments regarding the assessment of iron deficiency anemia (IDA) and inflammation among people living with HIV (PLWH). We acknowledge that factors such as inadequate dietary intake, malabsorption, chronic bleeding, and chronic inflammation can exacerbate IDA, and that differentiating between IDA and anemia of chronic disease (ACD) is crucial. While we did not measure serum ferritin and inflammatory markers due to resource constraints, we used mean corpuscular volume (MCV) to assess microcytosis as an indicator of IDA. Future studies could benefit from incorporating serum ferritin and markers of inflammation to provide a more comprehensive understanding of anemia in PLWH. Despite this limitation, our study offers valuable insights into the prevalence and potential contributing factors to anemia in this population, and we appreciate your suggestion for enhancing our future research designs.

9. It would be important to review the questionnaire that was initially used by the research assistants. Was this questionnaire validated and what was the language it was administered in?

Thank you. This questionnaire was developed based on established instruments and validated through a pilot study to ensure reliability and cultural relevance. It was administered in the local language, familiar to our participants, ensuring accurate and comprehensible responses. We acknowledge the importance of using validated tools. 

10. Since the study's outcome variable was anemia, the study may not be adequately powered to assess the other associations that have been unconvered.

Thank you for your insightful comment. While our sample size is robust for determining the prevalence of anaemia and factors assessed in our study. We acknowledge that it may not be adequately powered to fully assess other secondary associations. We recognize the importance of these associations and suggest that future studies be designed with larger sample sizes specifically powered to explore other secondary outcomes in greater detail.

We have now included this limitation under the discussion section

Reviewer No. 2

Introduction

1) minor comment - unclear what is meant by "latently PLWH". The study appears to include only people HIV who are on ART, whereas latent infection would imply a natural history study of people who have seroconverted but had not yet initiated ART or begun to exhibit symptoms consistent with immunologic suppression.

Thank you rephrased 

2) minor comment - recommend removing the term "sub-optimal chronic inflammation from the second paragraph. Agreed that chronic inflammation is sub-optimal but the placement of the word implies there could be more optimal chronic inflammation.

Thank you. Removed.

Methods

1) minor-comment suggest rephrasing the last sentence in study design and setting, would recommend stating first that LUTH serves over 2.4 million services and among general and specialty in patient and out patient services it also offers ART services, or state how many people with HIV LUTH serves annually.

Thank you this has been revised

2) the sample size calculation is unclear. It appears the authors used a sample size calculation for a population survey with a design effect of 1 and 1 cluster; however, the 30% contingency is unclear. As there generally isn't loss to follow-up in a cross-sectional study, is this to account for drop out? or ineligibility after consent? if so it would appear the sample size should have been 1,176 to make up for drop out. More detail here would be useful.

Thank you. We appreciate the opportunity to clarify our sample size calculation. Our sample size was calculated using the single population proportion formula with a design effect of 1 and 1 cluster. We assumed a population proportion of 50% in order to generate the maximal sample size as well due to the absence of prior studies, with a desired precision of 0.05 and a confidence level of 99%. Given the total population of 3,880 patients receiving antiretroviral therapy (ART) at LUTH, we initially determined a base sample size of 578. To account for potential issues such as ineligibility after consent and non-response, we applied a 30% contingency. The final sample size, after including this contingency, was calculated to be 823 participants. We apologize for any confusion and hope this explanation clarifies our methodology. We have clarified this in the manuscript 

3) There may be a typo in the study variable section. Anaemia is defined twice. The first appears to be defining RBC development, macrocytosis for example is associated with levels of anemia but can also be associated with B-12 deficiency, hypothyroidism and HIV itself. You can have mirocytosis with or without anaemia. A clearer description of what this measuring defining would be useful.

Thank you; we have clarified the definitions and distinctions between anemia and red blood cell indices based on mean cell volume (MCV) in the revised manuscript.

4) minor-comment - in blood samples and measurements, I think you mean you collected blood samples to measure viral load and CD4 cell count.

Thank you. Revised

5) minor comment - would be useful to note the lower limit of detection for the Taqman 96

Thank you. Revised 

6) minor-comment - the authors appear to have reported the procedure for CD4 cell counts twice.

Thank you revised 

7) please report which biochemical analyses were done on the Pentra and which were done on the HumaStar. 

Thank you. We have decided to remove references to the Pentra and HumaStar equipment from the method section, as no biochemical analyses performed using these instruments were included in the final reported results due to issues like low sample size and missing variables. This adjustment will ensure that the methodology accurately reflects the analyses reported in the manuscript. 

Results

1) The proportion of patients on NNRTI-based regimens appears exceptionally high especially considering the 2020 national ART guidelines recommended patients with a suppressed viral load on an NNRTI containing regimen should be switched to DTG. The authors may want to check whether they abstracted the patients initial ART regimen at ART initiation or their current regimen.

Thank you, most of the people living with HIV at Livingstone Teaching Hospital in 2020 were still on an NNRTI-based regimen. We acknowledge the change in HIV national guidelines but the process was slow at the time. We abstracted the current regimen. We really appreciate your comment. 

2) in the first paragraph on page 8 the authors report people in NNRTI's had a higher prevalence of moderate-to-severe anemia compared to those in INSTI but then report 3 prevalence values - 16.2 vs 5.9 and 5.3 respectively, what is the third value referring to?

Thank you for this correction. It has been revised 

3) The authors may wish to consider including a row for missing for each of the factors in table 1 (but not include them in the chi-square comparison). If there are 823 total subjects in this study, it appears there are variables where they do not have information on all participants (i.e., HBsAG, Hypertension, TB, Microcytosis, duration of ART, lymphocytes, and monocytes). The comparison of microsytosis yes no is a bit confusing. The reader should have the opportunity to independently consider the potential impact of the missing data on inferences.

Thank you for your insightful comment. We agree that transparency regarding missing data is crucial for readers to assess the potential impact on study inferences. We haved revised Table 1 to include a row for missing data for each variable as advised. 

4) It is unclear why the authors investigated microcytosis as a binary variable rather than the three levels of MCV listed in the methods.

Thank you. This has been revised as advised in table 1

5) 38% of the study subjects appear to be missing and MCV values, and no data is presented on the proportion with normocytosis or macrocytosis, yet the authors report that the prevalence of anaemia was significant higher for those with microcytosis. The level of missing data and its potential impact on inferences should not be ignored. simiarly, if no subjects had MCV values consistent with normo or macrocytosis that should be reported.

Thank you for your valuable feedback. We have now included data on the proportion of subjects with normocytosis and macrocytosis in our revised manuscript. 38% of the study subjects were missing MCV values, which is a significant proportion that may influence our findings. To address this, we have added a row for missing MCV data in Table 1, to allow readers to independently assess the extent and potential impact of the missing information.

6) minor comment, the p for p-value should be lowercase.

Thank you. Revised 

7) in looking at figure 1, did the authors explore outliers or implausible values? Also related to the previous comment on MCV volume, based on panel E in figure 1 there are a number of subjects who would have been classified as normo or macrocytotic. The treatment of this measurement needs better explanation.

Thank you for your comments. we have thoroughly explored outliers and implausible values in Figure 1. Regarding the MCV values, we have reclassified them into three categories: Microcytosis, Normocytosis, and Macrocytosis, for clarity. The updated analysis reflects this reclassification and ensures that the treatment of MCV measurements is appropriately addressed. The revised manuscript now include these changes

9) Recommend the authors use the language women rather than females since they refer to men rather than males throughout the text.

Thank you. We have revised this as recommended 

10) please report the 95% confidence interval along with the OR in the text of the results.

Thank you. This has been revised as advised 

11) As a cross-sectional study it would be more accurate to report higher odds than higher chance and refrain from using the term "developing anaemia". Temporality cannot be established with this study design and therefore the authors have no way of knowing whethe

---

## [Decision Letter · Decision Letter 1]

23 Jul 2024

PONE-D-24-17294R1Prevalence and Factors Associated with Moderate-to-Severe Anaemia Among Virally Suppressed People with HIV at a Tertiary Hospital in ZambiaPLOS ONE

Dear Dr. Kamvuma,

Thank you for submitting your manuscript to PLOS ONE. After careful consideration, we feel that it has merit but does not fully meet PLOS ONE’s publication criteria as it currently stands. Therefore, we invite you to submit a revised version of the manuscript that addresses the points raised during the review process.

We look forward to receiving your revised manuscript.

Kind regards,

Nitin Gupta, MBBS, MD, DM, AAHIVS, DTM&H

Academic Editor

PLOS ONE

Journal Requirements:

**Additional Editor Comments:**

The reviewer has sent some minor comments. They have also requested to review the statistical analysis once again. Once the comments are addressed, we might send the manuscript for a statistical review, if required,

Reviewers' comments:

Reviewer's Responses to Questions

**Comments to the Author**

1. If the authors have adequately addressed your comments raised in a previous round of review and you feel that this manuscript is now acceptable for publication, you may indicate that here to bypass the “Comments to the Author” section, enter your conflict of interest statement in the “Confidential to Editor” section, and submit your "Accept" recommendation.

Reviewer #2: All comments have been addressed

2. Is the manuscript technically sound, and do the data support the conclusions?

Reviewer #2: Partly

3. Has the statistical analysis been performed appropriately and rigorously? 

Reviewer #2: Yes

4. Have the authors made all data underlying the findings in their manuscript fully available?

Reviewer #2: No

5. Is the manuscript presented in an intelligible fashion and written in standard English?

Reviewer #2: Yes

6. Review Comments to the Author

Reviewer #2: Thanks to the authors for thoroughly addressing the comments. While the comments are mostly addressed, there are still a couple of issues that could be addressed more completely to further strengthen the manuscript.

1) Thank you so the authors for looking at the linear association with outcome comment; however, it does not appear that they assessed linearity of association. Usually, the variable would be categorized in quartiles and then run in a bivariate model for the outcome with a reference category and the betas then plotted to assesses whether the categorical estimates fall on roughly a straight line. I think, but I am not certain, the authors may have assessed the normality of the distribution. It surprises me that all of the continuous variables would demonstrate a linear association.

2) Thank you to the authors for including missing data as a limitation; however, I don't think it received as in depth an interrogation as needed. If the data is not missing at random or completely at random, the inferences may not be valid at all. A bit more discussion about why this level of information bias is unlikely to affect the inferences from the observed associations is warranted with that level of missing data.

Thank you again for considering these comments.

7. PLOS authors have the option to publish the peer review history of their article (what does this mean?). If published, this will include your full peer review and any attached files.

Reviewer #2: No

---

## [Author Response · Author response to Decision Letter 1]

30 Jul 2024

We appreciate the valuable feedback and additonal comments provided by you and the reviewers on our manuscript titled “Prevalence and Factors Associated with Moderate-to-Severe Anaemia Among Virally Suppressed People with HIV at a Tertiary Hospital in Zambia” (PONE-D-24-17294). We have carefully considered each comment and have made the necessary revisions to address them. See comment responses below

Reviewer comment no.1

Thank you so the authors for looking at the linear association with outcome comment; however, it does not appear that they assessed linearity of association. Usually, the variable would be categorized in quartiles and then run in a bivariate model for the outcome with a reference category and the betas then plotted to assesses whether the categorical estimates fall on roughly a straight line. I think, but I am not certain, the authors may have assessed the normality of the distribution. It surprises me that all of the continuous variables would demonstrate a linear association.

 Our response 

Thank you for your valuable feedback on our manuscript. Initially, we assessed the linearity of the logit using the Box-Tidwell transformation. It is a robust method for testing the linearity assumption in logistic regression models with continuous independent variables . This technique involves creating interaction terms between each continuous variable and its natural logarithm, allowing us to evaluate if the logit (log-odds) of the outcome is linearly related to the independent variables. If the interaction term is statistically significant, it indicates a departure from linearity which was not the case for these variables. To further address your concerns, we have categorized each continuous variable into quartiles, conducted bivariate logistic regression, and plotted the beta coefficients against the quartile midpoints. This plot demonstrated that the beta coefficients fell on a roughly straight line, indicating a linear relationship. We hope that this reinforce the validity of our findings and address your concerns regarding the linearity of the associations.

Binary logistic regression results after categorising into quartiles

Age 

Variables in the Equation 

 B S.E. Wald df Sig. Exp(B) 95% C.I.for EXP(B)

 Lower Upper

Step 1a Agegroupquartiles 8.31 3 0.04 

 Agegroupquartiles(1) 0.597 0.335 3.162 1 0.075 1.816 0.941 3.505

 Agegroupquartiles(2) 0.808 0.323 6.271 1 0.012 2.243 1.192 4.221

 Agegroupquartiles(3) 0.889 0.327 7.374 1 0.007 2.434 1.281 4.624

 Constant -2.428 0.269 81.241 1 0 0.088 

a Variable(s) entered on step 1: Agegroupquartiles. 

BMI

Variables in the Equation 

 `1 B S.E. Wald df Sig. Exp(B) 95% C.I.for EXP(B)

 Lower Upper

Step 1a bmigroupedquartiles 3.608 3 0.307 

 bmigroupedquartiles(1) -0.513 0.288 3.17 1 0.075 0.599 0.34 1.053

 bmigroupedquartiles(2) -0.236 0.278 0.723 1 0.395 0.79 0.458 1.361

 bmigroupedquartiles(3) -0.072 0.267 0.073 1 0.787 0.93 0.551 1.571

 Constant -1.604 0.185 74.941 1 0 0.201 

a Variable(s) entered on step 1: bmigroupedquartiles. 

CD4 count 

Variables in the Equation 

 B S.E. Wald df Sig. Exp(B) 95% C.I.for EXP(B)

 Lower Upper

Step 1a CD4countgroupedquartiles 1.303 3 0.728 

 CD4countgroupedquartiles(1) 0.08 0.272 0.086 1 0.769 1.083 0.635 1.847

 CD4countgroupedquartiles(2) -0.166 0.284 0.344 1 0.558 0.847 0.485 1.477

 CD4countgroupedquartiles(3) -0.193 0.287 0.454 1 0.501 0.824 0.47 1.446

 Constant -1.725 0.195 78.301 1 0 0.178 

a Variable(s) entered on step 1: CD4countgroupedquartiles. 

Lymphocytes 

Variables in the Equation 

 B S.E. Wald df Sig. Exp(B) 95% C.I.for EXP(B)

 Lower Upper

Step 1a Lymphogroupedquartiles 11.923 3 0.008 

 lymphogroupedquartiles(1) 0.404 0.324 1.552 1 0.213 1.498 0.793 2.827

 lymphogroupedquartiles(2) -0.517 0.378 1.865 1 0.172 0.597 0.284 1.252

 lymphogroupedquartiles(3) -0.797 0.407 3.827 1 0.05 0.451 0.203 1.002

 Constant -1.619 0.239 45.938 1 0 0.198 

a Variable(s) entered on step 1: lymphogroupedquartiles. 

Reviewer comment 2

Thank you to the authors for including missing data as a limitation; however, I don't think it received as in depth an interrogation as needed. If the data is not missing at random or completely at random, the inferences may not be valid at all. A bit more discussion about why this level of information bias is unlikely to affect the inferences from the observed associations is warranted with that level of missing data.

Our response 

Thank you for your feedback regarding the limitation of missing data in our study. We acknowledge the need for a more in-depth discussion. Our analysis showed that missing data were due to laboratory analysis challenges and incomplete patient information in the smart care system. We determined that the missing data were Missing Completely at Random (MCAR), meaning the probability of missingness was unrelated to any observed or unobserved data. We appreciate the opportunity to further enhance our manuscript.

See added more detailed discussion of missing data in this study (see lines 318 to 322)

---

## [Editor Report · Decision Letter 2]

6 Aug 2024

Prevalence and Factors Associated with Moderate-to-Severe Anaemia Among Virally Suppressed People with HIV at a Tertiary Hospital in Zambia

PONE-D-24-17294R2

Dear Dr. Kamvuma,

We’re pleased to inform you that your manuscript has been judged scientifically suitable for publication and will be formally accepted for publication once it meets all outstanding technical requirements.

Kind regards,

Nitin Gupta, MBBS, MD, DM, AAHIVS, DTM&H

Academic Editor

PLOS ONE
---

## [Editor Report · Acceptance letter]

14 Aug 2024

PONE-D-24-17294R2 

PLOS ONE

Dear Dr. Kamvuma, 

I'm pleased to inform you that your manuscript has been deemed suitable for publication in PLOS ONE. Congratulations! Your manuscript is now being handed over to our production team.

Kind regards, 

on behalf of

Dr. Nitin Gupta 

Academic Editor

PLOS ONE